# The Imbalance among Oxidative Biomarkers and Antioxidant Defense Systems in Thromboangiitis Obliterans (Winiwarter-Buerger Disease)

**DOI:** 10.3390/jcm9041036

**Published:** 2020-04-07

**Authors:** Hiva Sharebiani, Bahare Fazeli, Rosanna Maniscalco, Daniela Ligi, Ferdinando Mannello

**Affiliations:** 1Immunology Research Center, Inflammation and Inflammatory Diseases Division, School of Medicine, Mashhad University of Medical Sciences, Mashhad 9177948564, Iran; hivasharebiani@yahoo.com (H.S.); or bahar.fazeli@gmail.com (B.F.); 2Vascular Independent Research and Education, European Organization, 20157 Milan, Italy; 3Department of Biomolecular Sciences, Section of Biochemistry and Biotechnology, University “Carlo Bo” of Urbino, 61029 Urbino (PU), Italy; r.maniscalco@campus.uniurb.it (R.M.); daniela.ligi@uniurb.it (D.L.)

**Keywords:** Buerger’s disease, Thromboangiitis obliterans, oxidative stress, antioxidant capacity, myeloperoxidase, coenzyme Q10, superoxide dismutase, glutathione reductase, Malonyl-dialdehyde, protein carbonyl

## Abstract

(1) Background: Thromboangiitis obliterans or Winiwarter-Buerger disease (WBD), is an inflammatory, thrombotic occlusive, peripheral vascular disease, usually occurring in young smokers. The pathophysiological mechanisms underlying the disease are not clearly understood. The aim of this study is to investigate the imbalance between oxidants and antioxidants occurring in these patients. (2) Patients and Methods: In this cross-sectional study, 22 male patients with WBD and 20 healthy male smoking habit matched control group were included. To evaluate the possible sources of oxidative stress, the antioxidant biomarkers, and the markers of lipid peroxidation and protein oxidation, serum samples were analyzed for total oxidative status (TOS), total antioxidant capacity (TAC), myeloperoxidase (MPO), coenzyme Q10 (CoQ10), superoxide dismutase (SOD), glutathione reductase (GR), malondialdehyde (MDA), and protein carbonyl (PC) activity and/or content. (3) Results: The circulating levels of TOS, TAC, and CoQ10 were significantly higher in WBD patients, with respect to healthy smokers as controls. No significant difference was found among the serum level of PC, total cholesterol, MPO, and GR activity in WBD patients and healthy smoker controls. The activity of SOD and the mean serum level of MDA were significantly lower in WBD patients, with respect to healthy smoker controls. (4) Conclusion: Considerably high levels of oxidative stress were detected in WBD patients, which were greater than the antioxidant capacity. The low level of MDA may be associated with the enzymatic degradation of lipid peroxidation products. High levels of CoQ10 and low levels of SOD may be related to a harmful oxidative cooperation, leading to the vasoconstriction of WBD, representing a promising tool to discern possible different clinical risks of this poorly understood peripheral occlusive disease.

## 1. Introduction

Since the milestone landmark articles [1,2], thromboangiitis obliterans or Winiwarter-Buerger disease (WBD) has been defined as an inflammatory, thrombotic, occlusive, peripheral vascular disease, which usually occurs in young male smokers. The occlusion of small- and medium-sized vessels (arteries and veins of both upper and lower extremities) may lead to tissue or limb loss [3]. Although the clinic-pathological hallmarks, and some molecular mechanisms have been studied, the etiology, pathophysiology, and optimal therapy remain not yet fully defined or understood [4]. Although a significant relationship between smoking (both tobacco and cannabis products) and progression of WBD has been identified [5], smoking per se cannot explain the prevalence and distribution of this harmful vascular disease [6]. Despite efforts in previous years and advances in medical therapy [4,7,8], WBD patients may actually benefit only by blocking smoking habits (abstinence of all tobacco/cannabis product use) and through targeted endovascular therapy, effective for preserving limb loss/amputation, and useful for accelerating the healing process in Buerger’s ischemic ulcers [9].

Among the immunologic and inflammatory biomolecular hypotheses for the WBD disease, some biomarkers and biochemical mechanisms have been identified. Among them, crucial roles are played by cytokines and chemokines [10,11,12], adhesion molecules [13,14,15], *Rickettsia rickettsii* and *Porphyromonas gingivalis* (through toll-like receptors) [16,17,18,19], angiogenic factors [20], catecholamines [21], inflammation on sympathetic ganglia [22], T cells/macrophages/dendritic cells (intima infiltration of vessels) [23], accumulation of immunoglobulins, immune complexes and complement factors on sub-endothelial elastic lamina [24], urinary cotinine [25], circulating auto-antibodies [26], heme oxygenase 1 and the inducible isozyme of nitric oxide synthase [27], and matrix metalloproteinases [28] (as reviewed in [29]).

Interestingly, clinical and biological studies indicate that oxidative stress may significantly contribute to the initiation and progression of WBD. In particular, it has been demonstrated that in WBD patients there is a significantly altered pro-oxidant/antioxidant imbalance, with respect to the healthy controls [30,31]. However, these studies suggest that, besides smoking, other biomolecular mechanisms may be involved and responsible for the huge amount of oxidative stress found in WBD patients, even if the effects of smoking could amplify the impairment of oxidative–antioxidative pathways in WBD patients, leading to both the inflammatory and thrombotic events of Buerger’s disease.

In this interesting context, the aim of our cross-sectional study is to investigate further unexplored biomarkers, identify cell sources and biochemical pathways linked to oxidative stress in WBD patients, and compare data with those detected in healthy smokers as the control group. For this reason, the serum levels of total oxidative status (TOS) and total antioxidant capacity (TAC) (also known as non-enzymatic anti-oxidant capacity [32]) were measured in the WBD patients and healthy smoker controls. Furthermore, we analyzed serum levels of myeloperoxidase (MPO), an enzyme produced/secreted by neutrophils and monocytes during oxidative stress in smokers, and involved in both cardiovascular and lung diseases [33,34]. In addition, the serum level of Coenzyme Q10 (CoQ10), a lipophilic endogenously synthesized antioxidant playing a crucial role in mitochondrial energy production, postulated to be actively degraded/reduced by smoking, and involved in mitochondrial dysfunction [35], was also evaluated. Finally, serum enzymatic activities of superoxide dismutase (SOD) and glutathione reductase (GSH-Red), well known antioxidant enzymes counteracting free radical disturbances and protecting cell and mitochondrial membranes [36], were tested in conjunction with the molecular alteration of lipids and proteins through the release in blood of malondialdehyde (MDA) and protein carbonyl (PC) [37,38], well known biomarkers of free-radical-mediated lipid peroxidation and protein oxidation, respectively.

## 2. Patients and Methods

In this cross-sectional study, 22 male patients with WBD (diagnosed according to Shionoya’s criteria) [39,40] and 20 healthy male smokers (as a control group) were included. Patients’ written consent, demographic characteristics, and clinical manifestations were obtained during admission to the Buerger’s Disease Clinic (Mashhad University of Medical Sciences, Iran) from December 2017 to December 2018, and were maintained in their medical records (Ethical code: MUMS-961484).

Our series of WBD patients and healthy subjects resembled each other for general clinical and biochemical characteristics (no statistically significant differences for cigarette smoking; albumin, globulins, creatinine, alanine transaminase, aspartate transaminase, and homocysteine levels; blood pressure; and body mass index). None of the subjects had taken antioxidant vitamins and/or drugs that may have induced possible bias for either the generation of reactive oxygen species or their inhibition/chelation/down-regulation.

After 8–10 h fasting and smoke abstinence, blood samples were drawn into glass tubes containing ethylenediaminetetraacetic acid (EDTA), but not containing any anticoagulant. After allowing the serum to clot for 5–10 min at room temperature, plasma samples (containing EDTA) and serum (without anticoagulant) were obtained by centrifugation at 1000× *g* for 20 min and stored at −80 °C until analysis. The supernatants were carefully collected and, when sediments occurred during storage, centrifugation was performed again.

Serum total cholesterol and routine biomarkers level were determined on a Modular 7180 discrete clinical AutoAnalyzer (Hitachi) using a commercially available diagnostics kit. The other biochemical parameters were determined through the DU-series spectrophotometer (Beckman) according to the manufacturer’s instructions of the diagnostic kits.

Protein concentrations were determined by the Lowry method, with bovine serum albumin as the standard. For all parameters, standard curves were established with a set of serial dilutions of the samples; the resulting equations were used to calculate the unknown sample concentrations and to evaluate the linearity of the test, according to the manufacturer’s instructions.

The serum levels of total oxidative status and total antioxidant capacity were measured using the ELISA kit (ZellBio, GmbH, Germany, ZB-TOS-96A and ZB-TAC-96A).

In detail, the Total Antioxidant Capacity (TAC) Assay Kit measured either the combination of small molecule antioxidants and proteins or small molecules alone. Cu^2+^ ion is converted to Cu^+^ by both small molecules and protein. The TAC antioxidant mix prevents Cu^2+^ reduction by protein, enabling the analysis of only the small molecule antioxidants. The reduced Cu^+^ ion is chelated with a colorimetric probe giving a broad absorbance peak around 570 nm, proportional to the total antioxidant capacity. According to manufacturer’s instructions, we used Trolox to standardize antioxidants, being that all other antioxidants were measured in Trolox equivalents. We dissolved the lyophilized Trolox standard in 20 µL of pure DMSO by vortexing, then added 980 µl of distilled water and mixed well, generating a 1 mM solution. Following reconstitution, aliquots were stored at −20 °C. The reconstituted standard was stable for 4 months, stored at −20 °C. No sample purification from blood sample was necessary; serum samples (usually 0.01–0.1 µL) were assayed without dilution. After adding 100 µL Cu^2+^ of working solution to all standard and sample wells, the plates were covered and incubated at room temperature for 1.5 h and then were read the absorbance at 570 nm using the plate reader, as a function of Trolox concentration. The absorbance of samples was in the linear range of the standard curve (0–20 nmol/well). If they were outside of this range, they were rediluted in distilled water and re-ran again. The detection limit of the assay was approximately 0.1 nmol per well.

Concerning the Total Oxidative Status (TOS) assay kit, the serum TOS levels were determined using an automated measurement method described elsewhere [41]. Briefly, oxidants present in the sample oxidize the ferrous ion-o-dianisidine (Fe^2+^/3-3′-dimethoxybenzidine) complexes into ferric ions (Fe^3+^). The oxidation reaction is enhanced by glycerol molecules that are abundantly present in the reaction medium. The final reagent was composed of 150 μM xylenol orange, 140 mM NaCl, and 1.35 M glycerol. The pH value of the reagent was 1.75. The reactive reagent was composed of 5 mM ferrous ammonium sulfate and 10 mM o-dianisidine dihydrochloride. The ferric ions formed a colored complex with xylenol orange [o-cresosulfonphthalein-3,3-bis(sodium methyliminodiacetate)] in an acidic medium. The wavelength was set at 560 nm and the reading of the linear assay/calibration was an end-point measurement after 5 min after the reaction trace draws a plateau line. Therefore, the color intensity, spectrophotometrically measured, is related to the total number of oxidant molecules present in the sample. The assay was calibrated with hydrogen peroxide and the results were expressed in terms of micromolar hydrogen peroxide equivalent per liter (μmol H_2_O_2_ equivalent/L). The lower detection limit was 1.13 μmol H_2_O_2_ equivalent/L.

Coenzyme Q10 (Ubiquinone-10, CAS 303-98-0), malondialdehyde, and protein carbonyl were evaluated using the ELISA kit (ZellBio, GmbH, Germany, ZB-CoQ10-96A, ZB-TMD-96A, and ZB-PC-96A). In detail, the Coenzyme Q10 (CoQ10) assay kit employed the competitive inhibition enzyme immunoassay technique. The microtiter plate was pre-coated with an antibody specific to CoQ10. Standards or samples were added to the appropriate microtiter plate wells with horseradish peroxidase-conjugated CoQ10. The competitive inhibition reaction was launched between with HRP-conjugated CoQ10 and CoQ10 in samples. The peroxidase substrate solution (tetramethylbenzidine) was added to the wells and the color develops in opposite to the amount of CoQ10 in the sample. The color development was stopped and the intensity of the color was measured. According to the manufacturer’s instructions, the serum or plasma samples were diluted with kit sample diluent (1:100). For measuring absorbance, the microplate reader was set at 450 nm. The minimum detectable amount of human CoQ10 was 1.56 ng/mL.

Concerning the malondialdehyde (MDA) assay kit, we used a quantitative colorimetric method on the basis of malondialdehyde/Thiobarbituric Acid Reactive Substances (TBARS) assay, a well standardized tool for assessment of lipid peroxidation in biological samples [42]. The chemical adduct formed by the reaction of MDA and thiobarbituric acid (TBA) was colorimetrically measured in acidic media; and after heating (90–100 °C), analyzed the absorbance at 535 nm. According to the manufacturer’s instruction, MDA levels in unknown samples were calculated based on a standard curve, which were drawn using standard point absorbance. The MDA assay kit can determine MDA in biological samples with 0.1 µM sensitivity.

Finally, protein carbonyls (PC) were assayed according to the well-known and previously detailed method [43]. The most common products of early protein oxidation in biological samples are the protein carbonyl derivatives (especially from Pro, Arg, Lys, and Thr amino acid residues). These derivatives are chemically stable and serve as markers of oxidative stress for most types of ROS. Many of the current assays involve derivatization of the carbonyl group with dinitrophenylhydrazine. Briefly, after the incubation of equal volumes of plasma and 2,4-dinitrophenylhydrazine at 50 °C for 1 h, proteins were precipitated with 20% (*v*:*v*) of trichloroacetic acid and the unreacted dye was removed by centrifugation. The pellet was dissolved in 1 M NaOH, and the formation of a Schiff base producing the corresponding hydrazone was analyzed spectrophotometrically at 375 nm. According to manufacturer’s instruction, PC levels in unknown samples were calculated based on the standard curve, which was drawn using standard point absorbance. The curve was obtained by preparing 10 μg/mL of reduced or oxidized BSA, by diluting the 1 mg/mL BSA standards in phosphate buffered saline.

In addition, the enzyme levels of superoxide dismutase (EC 1.15.1.1) and glutathione reductase (EC 1.8.1.7) were measured using the ELISA method (ZellBio, GmbH, Germany, ZB-SOD-96A and ZB-GR-96A). The level of myeloperoxidase (EC 1.11.2.2) was also analyzed using the ELISA method (eBioscience, BMS2038INST).

Superoxide dismutase (SOD)(EC 1.15.1.1) activity was determined for assaying both intra- and extracellular enzymes (MnSOD and Cu-ZnSOD, respectively), according to the original method previously described [44]. This assay activity involves inhibition of nitro blue tetrazolium reduction, with xanthine–xanthine oxidase used as a superoxide generator. Briefly, the activity was assessed in the ethanol phase of the sample after 1.0 mL ethanol/chloroform mixture (5/3, *v*/*v*) was added to the same volume of sample and centrifuged. One unit of SOD was defined as the enzyme amount causing 50% inhibition in the NBT reduction rate, recording, spectrophotometrically, the absorbance change at 470 nm over 20 min at 25 °C, as fully described in the well standardized method [45].

Glutathione Reductase (EC 1.8.1.7)(GR) catalyzes the NADPH-dependent reduction of oxidized glutathione (GSSG) to reduced glutathione (GSH), which plays an important role in the GSH redox cycle that maintains adequate levels of reduced GSH, being high GSH/GSSG ratio essential for protection against oxidative stress. Briefly, the GR assay kit was based on a colorimetric assay for measuring GR reduction of GSSG to GSH, which reacts with 5, 5′-Dithiobis (2-nitrobenzoic acid) (DTNB) to generate TNB2- (measurable at 412 nm as yellow color) [46]. Undiluted blood samples were analyzed against a standard curve of TNB. The assay shows a sensitivity >0.1 U/L.

Finally, Myeloperoxidase (MPO)(EC 1.11.2.2) was based on the sandwich ELISA principle. Each well of the supplied microtiter plate has been pre-coated with a target specific capture antibody. MPO standards or presents in blood samples (usually 100-fold diluted in order for their optical density (OD) readings to fall within the Standard Curve) were captured (as antigen) by specific antibody and revealed by biotin-conjugated detection. An Avidin-Horseradish Peroxidase conjugate binds to the biotin allowing the substrate (3,3′,5,5′-tetramethylbenzidine) to react with the HRP enzyme, resulting in color development. A sulfuric acid stop solution was added to terminate color development reaction and then the optical density (OD) of the well measured at a wavelength of 450 nm. The optical density of an unknown sample can then be compared to a standard curve, generated using known antigen concentrations in order to determine its antigen concentration. The detection range was 1.56–100 ng/mL, with a sensitivity of 0.65 ng/mL.

All reagents, unless otherwise noted, were obtained from Sigma-Aldrich, Milan, Iatly.

All data, expressed as mean ± standard deviation (SD), were analyzed using Statistical Package for the Social Sciences (SPSS, IBM, Milan, Italy) version 16.0, setting significant level at *p* < 0.05; all data were normally distributed and underwent equal variance testing. Statistical analysis was performed by the Mann–Whitney U-test and analysis of variance with Tukey’s posttest. Decision trees were designed and evaluated using Rapid Miner software V5.3.

## 3. Results

In this study, 22 male WBD patients with a mean age of 40.6 ± 1.1 years and 20 male smokers with mean age of 42 ± 1.4 (as control group) were included. No significant difference between the ages of both groups was found (*p* = 0.33).

According to patient self-reports, the mean number of cigarettes smoked per day was 17.7 ± 1.2 and 18.9 ± 3.3, in the patient and control group, respectively. No significant difference in terms of cigarette consumption was found among the two groups (*p* = 0.85).

All of the biochemical data are summarized in Table 1.

### 3.1. Serum Levels of Total Oxidative Stress (TOS) and Total Antioxidant Capacity (TAC)

TOS was significantly higher in WBD patients compared to the controls (*p* = 0.007). Similarly, the mean serum TAC levels were significantly increased in WBD patients compared to the healthy smoker controls (*p* = 0.04). Additionally, the TOS–TAC ratio was significantly higher (*p* < 0.001) in WBD patients than that found in the healthy control group.

### 3.2. Serum Levels of Myeloperoxidase (MPO), Glutathione Reductase (GR), and Superoxide Dismutase (SOD) Activity

The serum levels of MPO in WBD patients showed no significant difference between patients and healthy smokers as controls (*p* = 0.49).

Comparing WBD patients and healthy smokers, the mean levels of GR activity were found not statistically different (*p* = 0.22).

WBD patients showed significantly lower levels of SOD enzyme activity than that found in healthy smokers (*p* = 0.002), even if no difference was found for cigarette smoking habits between the two groups (*p* = 0.85).

### 3.3. Serum Levels of Coenzyme Q10 (CoQ10) and Cholesterol

The serum level of CoQ10 was found significantly higher in WBD patients than healthy smokers (*p* < 0.001).

Notably, no significant difference in terms of total cholesterol was found between the WBD patients and the healthy smokers as control group (*p* = 0.57).

### 3.4. Biomarkers of Lipid Peroxidation and Protein Oxidation

The mean serum level of MDA was found significantly lower in WBD patients, with respect to controls (*p* < 0.001). On the contrary, no statistically significant difference in the mean serum level of PC was found in our series of patients (*p* = 0.83).

The WBD patient group and healthy smokers’ group were classified according to the decision trees on the basis of SOD, CoQ10, MDA, and PC (i.e., the biomarkers that showed significant difference among patients) (Figure 1A,B).

As depicted in both decision trees, the common root is represented by the enzyme SOD; therefore, by our series of analyses, SOD may be recognized as the most important classification marker in Winiwarter-Buerger disease, as depicted in Figure 1 and statistically evaluated in Table 2.

## 4. Discussions

Winiwarter-Buerger disease (WBD) or thromboangiitis obliterans has been clearly defined as a serious medical and social problem, characterized as a nonatherosclerotic, inflammatory vasculitis strongly associated with smoking and affecting vessels of both upper and lower extremities [4,47].

WBD is characterized as a nonatherosclerotic thrombotic occlusive peripheral vascular disease, affecting arteries and veins with the presence of highly inflammatory thrombus [47,48,49]. Although strongly associated to the use of tobacco and cannabis products use, per se, are not justifying the well described oxidative stress responsible for the progression and worsening of this disease, sadly suggesting that little progress has been made in the understanding of its pathophysiology and treatment [4,8,50].

Although several studies have focused on discern the molecular mechanisms involved in the etiopathogenesis and pathophysiology pathways (as reviewed in [51,52]), and according to some studies performed to evaluate different oxidative and antioxidant pathways in Winiwarter-Buerger disease [30,31,53,54], to our knowledge, the present study is the first report demonstrating a possible link between SOD and MDA down-regulations, related also to a significant increase of CoQ10 levels in WBD, with respect to healthy smokers.

Despite the studies on pathophysiology (reviewed in [29,51], actually, the detailed etiopathogenesis and molecular mechanisms are still unknown. Among the possible biomolecular hypotheses [10,12,14,18,20,21,22,23,27,50,54,55,56], oxidative stress has been crucially indicated as an important factor for both the initiation and progression of WBD [30,31,51,54].

According to our previous results [31] and literature data [30] (reviewed in [51]), in WBD there is an evident and significant impairment of the balance between oxidative stress and the anti-oxidant capacity. It has been detected that pro-oxidant/antioxidant balance (PAB) significantly increased in WBD patients when compared with both the smokers and non-smokers groups [31], suggesting significant roles and effects of cigarette smoking to induce ROS release by neutrophils and monocytes, paving the way for an increasing oxidative stress microenvironment, driving to both inflammatory and thrombotic events typical of WBD. Our results are in full agreement with the previous findings, revealing significantly higher levels of total oxidative status (TOS) in WBD patients, even if not linked and/or limited only to smoking habits.

In this respect, it has been widely demonstrated that both endothelial and leukocyte cells (e.g., neutrophils and macrophages) under pro-inflammatory stimuli may both produce and scavenge ROS during physiological and pathological conditions; ROS can also be generated during both intracellular arachidonic acid metabolism and in mitochondria [57]. Accordingly, enzymatic and non-enzymatic-based antioxidant defense systems may protect the cells from the oxidative stress-induced damages [48]. In agreement with literature data, our results of significantly higher levels of TOS, TAC, and their ratios in WBD patients, with respect to smoker healthy controls, suggest that smoking itself is not, per se, the only thing responsible for inducing high oxidative stress and altered antioxidant response. In fact, besides smoking habits, the increased susceptibility to develop WBD might be also due to the presence of a mutation of the T-786C eNOS gene observed in WBD patients, which, through the regulation of nitric oxide bioavailability, may impair the oxidative unbalance caused by cigarette smoke [58]. Accordingly, it has been recently hypothesized that in WBD patients, oxidative stress may originate through the NF-kB/iNOS-NO pathway, able to interact with ROS modifying cells and mitochondria membrane lipids, inducing endothelial dysfunction [51]. In agreement with the hypothesis explaining our evidence of the reduction of SOD in WDB patients as a possible factor mitigating the bioavailability of NO (by the following pathway O2^−^ + NO -> ONOO), it is noteworthy that the downstream oxidation pathway typically involves nitration of tyrosine residues to form nitro-tyrosine; this could be measured in the serum of patients to determine if NO was consumed by O2^−^, as previously suggested in WBD patients [30,51,54].

Reactive oxygen species can trigger all types of cell biomolecules and, as highlighted and depicted in WBD [51], superoxide may act as scavengers of nitric oxide, forming peroxynitrite-dependent oxidative stress and leading to vascular dysfunction. In this respect, superoxide dismutase (SOD) play, as a part of the mitochondrial structure, a crucial role among the antioxidant defenses, minimizing the cellular damage by oxidative stressors. This enzyme also participates in vascular tonicity via the regulation and bioactivity of nitrite oxide levels by the endothelium; in fact, controlling the amount of superoxide anions by SOD is critical for preserving NO bioactivity in the vessel wall [59]. Our data of significantly lower serum levels of SOD in WBD patients than healthy smokers is in agreement to previously reported results of lower levels of intra-erythrocyte SOD in WBD, with respect to peripheral arterial occlusive disease [30,53]. Our results emphasize literature data, suggesting that in the presence of low-activity of SOD (both intra-erythrocyte and extra-cellular low levels), the reaction between superoxide anions and NO will lead to a loss of NO bioactivity and consequently vasoconstriction, which is one of the main manifestations in Winiwarter-Buerger disease [4,47]. Moreover, it has been demonstrated that the degree of tissue hypoxia, due to alterations of the antioxidant systems of the blood and nitrogen oxide, may critically affects occlusive diseases, including WBD [53].

Notably, during WBD progression, it has been demonstrated that oxygen free radical and lipid peroxide were markedly increased in conjunction to lower levels of erythrocyte SOD; this unbalance may actively participate in vascular endothelial cell injury and the detection of these substances might provide complementary evidence for syndrome differentiation of WBD [60].

Noteworthy, lower levels of NO in WBD patients compared to smokers, and also lower levels of NO in WBD patients with below-knee amputations compared to non-amputees, has also been demonstrated [54], supporting the crucial role of SOD as possible primary enzymatic defect in WBD and supporting the hypothesis about the measurement of SOD activity as a useful parameter for the differential diagnosis of Winiwarter-Buerger disease (as depicted and analyzed in our decision tree models) (Figure 1 and Table 2).

Previous studies reported that different biomarkers of oxidative stress are increased in patients affected by peripheral artery diseases, both due to the increased rate of ROS formation, or to a decreased clearance by antioxidant mechanisms. Interestingly, only one study was found comparing oxidative stress and antioxidant defenses in WBD vs. atherosclerotic peripheral arterial occlusive disease, reporting that the oxidative balance is more seriously impaired in WBD patients [30,51]. Additionally, we confirmed that the level of oxidants is greater in WBD patients than the antioxidant capacity in healthy smokers and non-smokers, supporting/strengthening the hypothesis of both a biochemically enhanced and clinically impairment of oxidant/antioxidant balance in Winiwarter-Buerger disease [8,51]. Interestingly, the production of ROS from intracellular and extracellular sources may affect both endothelial cell morphology and function; thus, also promoting the expression of adhesion molecules, which may represent pro-inflammatory signals and increase the risk of thrombosis [61].

It is well known and characterized that ROS are able to induce modifications in all macromolecules, in particular on lipids (through the mechanism of lipid peroxidation) and proteins (by time-dependent protein oxidation). In this respect, two main biomarkers (malondialdehyde, MDA; and protein carbonyl, PC) have been utilized to identify, characterize, and follow the oxidation initiation and progression of both lipids and proteins, respectively [62].

MDA has been widely used for many years as a convenient biomarker of lipid peroxidation, identified as an end-product generated by the decomposition of arachidonic acid and larger polyunsaturated fatty acids through enzymatic (biosynthesis of thromboxane A2 in platelet) or non-enzymatic (lipid peroxidation) processes. MDA may also be additionally eliminated through the formation of MDA-protein and/or MDA-DNA adducts (inducing also cell death) and degraded by the enzymatic modifications (mainly via aldehyde dehydrogenase and decarboxylase) [63]. In our present study, MDA was found significantly decreased in WBD patients compared to the healthy smoker controls. Low levels of MDA in our Winiwarter-Buerger patients might be due to further enzymatic MDA catabolism; our data suggest that enlarged mitochondria (as a consequence of high levels of CoQ10 in the WBD group) [64] may significantly increase the MDA enzymatic catabolism and degradation [63] and, consequently, show lower levels of MDA in the WBD patients compared to the healthy smoker controls. Further studies are in itinere to evaluate if MDA enzymatic catabolism are time-dependent or linked to progression of WBD.

Interestingly, MDA plays a role, also, in angiogenesis by influencing both vascular endothelial growth factor (VEGF) expression and NO alteration [65]; therefore, our data of significantly decreased serum levels of MDA might hypothetically take part in disturbing the process of angiogenesis typical of WBD [20,66,67].

Since MDA levels were unexpectedly low in TAO patients, we also evaluated the protein oxidation profile, analyzing the blood levels of protein carbonyl (PC), which has been demonstrated as a marker of protein oxidation [68]. Our results appear in agreement with the literature evidence showing that oxidizing species derived from the activity of myeloperoxidase (MPO) may lead to the formation of carbonyl groups in proteins with less modification in lipids [69]. However, the serum level of PC had no significant difference between our series of both patients and controls. To confirm our results, we evaluated the serum level of MPO, and no significant difference between the patients and controls was found. Further studies are in itinere to discern the complex and intricate network among oxidative stress and proteins/lipids in WBD patients.

As our results indicate, PC level has no significant difference between WBD patients and smoker controls; this clue could lead to the following different types of protein oxidative modifications (e.g., early protein carbonyls (PC) and late advanced oxidative protein products (AOPP)). This approach (in itinere in our labs) could help to understand what type of ROS and/or RNS is involved during the oxidative process and if these pathways may be time-dependent or smoke-dependent, providing useful suggestions for WBD affected patients’ identification/characterization. According to these hypotheses, and in agreement to our evidence of low activity of SOD in WBD patients, the presence of 3-nitrotyrosine could indicate the oxidative damage mediated by peroxynitrite, but not counteracted by adequate activity of SOD; this will set the basis for our, in itinere, next studies.

High levels of oxidative stress and related inflammation are also associated with mitochondrial dysfunction [70], and were found significantly higher in patients affected by peripheral arterial diseases [71]. No data are available about direct or indirect involvement of mitochondria dysfunction in WBD, even though it might represent a further explanation for the considerable oxidative stress in these patients. Our findings about CoQ10 may provide some clues in support of this hypothesis. In fact, in agreement with the literature knowledge [53], we reported the significantly higher levels of CoQ10 antioxidant activity in WBD patients, with respect to smoker healthy controls, suggesting that high levels of CoQ10 may represent an essential component of the mitochondrial membrane and respiratory chain replacing mitochondrial dynamics, protecting and restoring cells from oxidative stimuli, and able to ameliorate mitochondrial dysfunction in different in vitro and animal models [64,72,73,74].

Although CoQ10 molecular mechanisms and pathways could act as a “double-edged sword” (especially in the presence of mitochondrial dysfunction) [75], our data of significantly higher levels of CoQ10 in WBD patients, with respect to healthy smoker controls, could represent the marker of increase mitochondrial dynamics. Our data are the first evidence of high CoQ10 in WBD and might be related to lower levels of mitophagy, proning the WBD patients to the inability of clearing damaged mitochondria containing high amounts of superoxide ions [76]. In addition, CoQ10 may act also as an antiangiogenic factor through the inhibition of both fibroblast and endothelial growth factors expression, associated with a decreased ability of restoring morphology and functions of endothelial cells [77]. In this respect, oxidative stress and ROS have been proposed as pro-angiogenic factors in vascular disorders [78]. They act both directly and by generating pro-angiogenic oxidation products, mainly through HIF/VEGF signaling [79]. It is important to note that angiogenesis is a process dysregulated in WBD, and it has been demonstrated that long-term treatment with angiogenic medications for WBD in different geographic areas are highly recommended [8,66]. In this respect, angiogenesis is an efficient mechanism in WBD, and could be compromised by diminished SOD activity, mainly due to the possibility of SOD-derived H_2_O_2_ to induce the VEGF signaling pathway [59,66,80]. Noteworthy, it has been previously reported that VEGF signaling is altered in WBD patients, due to increased expression of specific receptor VEGF-R1 (known as an angiogenesis inhibitor) in the WBD patients and smokers vs. non-smoker subjects [10].

Therefore, more studies are needed to strengthen and support the present hypothesis that the inhibition of mitophagy by high levels of CoQ10 can induce severe oxidative stress in the context of mitochondrial dysfunction status in WBD patients, with respect to smoker healthy controls and non-smokers.

A review has discussed several of the limitations associated with pan TAC assays and, most importantly, that TAC assays exclude endogenous enzymatic activities of antioxidant systems, including catalase, and glutathione [32]. As correctly discussed, the TAC assay is not truly indicative of the “total” antioxidant capacity, but rather the ‘non-enzymatic antioxidant capacity (NEAC). Under this view, further explanation of our results may be related to endogenous antioxidant systems (such as catalase), which could explain why protein carbonyl levels did not raise, and could also explain the decrease in the oxidant marker MDA. Although a previous study demonstrated that in WDB patients the erythrocyte levels of GR, SOD, and CAT were significantly low [30], the hypothesis that oxidative stress, per se, does not lead to oxidative damage due to compensatory regulatory mechanisms needs to be confirmed and furtherly analyzed, especially in WBD patients.

Finally, it has been recently demonstrated that SOD activity is higher in women than in men [81], which could help explain why men tend to be more common victims of Winiwarter-Buerger disease [4,47,66].

In all, according to the data mining of our study, and being aware that more focused studies are needed to strength our results and support this hypothesis, our evidence showing significantly high oxidative levels, high amounts of CoQ10, and low SOD anti-oxidant activity in WBD might pave the way for more focused studies on SOD activity, as a primary key enzyme responsible for initiation/progression of WBD through vascular dysfunction, vasoconstriction, and impaired angiogenesis linked to impaired oxidant/antioxidant balance.

## 5. Conclusions

Taken together, considerably high levels of oxidative stress were detected in WBD, with respect to healthy smokers as control. The high amount of total oxidative species was found significantly greater than the antioxidant defense capacity, increasing, significantly, the ratio of TOS/TAC in Winiwarter-Buerger patients. The characteristic imbalance among oxidative biomarkers and antioxidant defense system that we found in WBD patients was depicted and summarized in Figure 2.

The low level of MDA in WBD patients may be linked to crucial further enzymatic catabolism of this lipid peroxidation product, which may induce down-regulation of some growth factors involved in angiogenesis. Moreover, high levels of CoQ10 and low levels of SOD and MDA may explain the insufficient angiogenesis in WBD.

According to our results, it seems possible that there is a primary enzymatic disorder of SOD activity, which, through its down-regulation, could lead to severe oxidative stress (increasing reactive species of both oxygen and nitrogen) and enhance vasoconstriction, worsening WBD. 

Significantly higher levels of CoQ10 in WBD seem like a defense mechanism against high oxidative stress as they could increase mitochondrial biogenesis (probably reducing mitophagy). However, in patients with morphological and molecular mitochondrial dysfunctions (e.g., SOD polymorphisms), enlarged mitochondria could be linked to significantly higher oxidative stress. Further studies on SOD genetic and epigenetic modification in WBD may be highly recommended as a possible biomarker and target for future treatments.

Considering the current literature and our data we hypothesize that SOD might be an interesting biomarker for risk assessment in patients with WBD. The combination of classical risk scores with biomarkers, such as SOD, MDA, and CoQ10 might be an innovative promising approach in preventive clinical medicine of a peripheral vascular disease not fully understood.

## Figures and Tables

**Figure 1 jcm-09-01036-f001:**
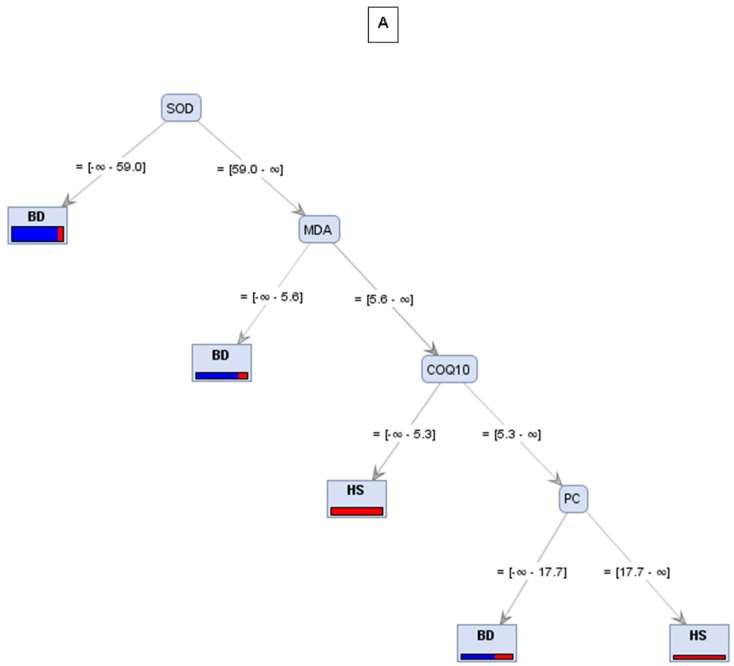
Predictive modeling approaches of Buerger’s disease (BD) and healthy smoker (HS) based on biomarkers analyzed in blood samples. **A**: decision tree linking SOD to lipid (MDA) and protein (PC) oxidation markers. **B**: the predictive values relating SOD to antioxidant mitochondrial markers CoQ-10 B.

**Figure 2 jcm-09-01036-f002:**
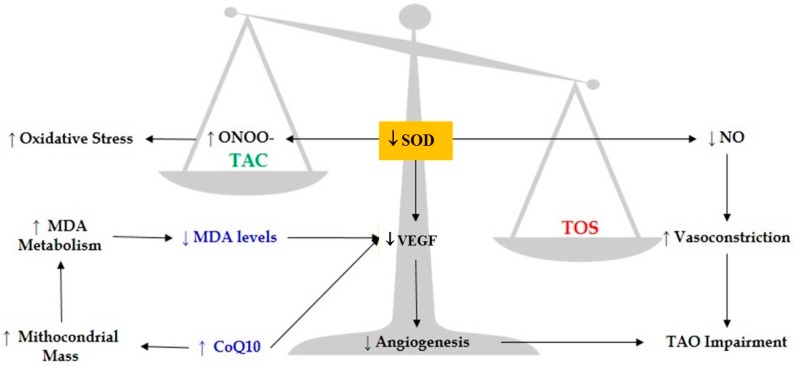
Schematic representation of the imbalance between oxidative stress/antioxidant system and the biomolecular cascade of events in Winiwarter-Buerger patients.

**Table 1 jcm-09-01036-t001:** Serum levels of soluble biomarkers in patients with Winiwarter-Buerger disease vs. smokers as control group.

	Buerger’s Patients	Smoker Controls	*p* Value
Total Oxidative Stress (TOS, μM)	2.12 ± 0.5	0.06 ± 0.006	0.007
Total Antioxidant Capacity (TAC, nM)	0.32 ± 0.02	0.2 ± 0.05	0.04
TOS-TAC Ratio	6.4 ± 1.6	1.7 ± 1.2	<0.001
Myeloperoxidase (MPO, ng/mL)	1.55 ± 0.61	1.42 ± 0.42	0.49
Malondialdehyde (MDA, μM)	5.3 ± 2.8	13 ± 6.2	<0.001
Superoxide Dismutase (SOD, U/L)	52.1 ± 8.53	79.6 ± 31.8	0.002
Glutathione Reductase (GR, U/L)	43.5 ± 22.8	35.24 ± 25.1	0.22
Protein Carbonyl (PC, ng/mL)	22.4 ± 16.2	24.9 ± 16.2	0.83
Coenzyme Q10 (CoQ10, pg/L)	5.5 ± 1.2	3.7 ± 2.5	<0.001
Total Cholesterol (mg/dl)	157.5 ± 35	193 ± 40	0.57

**Table 2 jcm-09-01036-t002:** Statistical evaluation of decision tree models.

Trees	Accuracy	Precision	Recall
A	76.92%	77.78%	50%
B	76.92%	85.71%	42.86%

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
