# Peer review of "The Imbalance among Oxidative Biomarkers and Antioxidant Defense Systems in Thromboangiitis Obliterans (Winiwarter-Buerger Disease)"

_jcm, 2020, doi:10.3390/jcm9041036_

Round 1
Reviewer 1 Report
The paper concerns an interesting topic related to the Burger disease patogenesis. The Authors compared the results achieved in 22 male subjects and 20 healthy patients - an investigation of various paramateres concerning oxidative stress and antioxidant mechanism was performed as described in the methods. The results are interesting and well discussed in the discussion part. Some question and remarks from the reviewer:
- the table 1 data are the repetition of the data presented in the text above ( the same values as described in the text description) - I think You should decide in which form Your present the results - the text descritpion or in the table as this concerns the same.
- most of the patienst were smokers - when the blood for the test was taken - in the morning? after the cigarete smoking ? before? - the condition of the blood taking in the protocol should be specified
- in the "conclusion" part of the text You have refferences? I think this part of conclusion should be moved to the discussion summary as this concerns the literature discusssion and You should based Your conclusion on Your research (so please modify the "conclusion")
Author Response
REVIEWER #1:
COMMENT 1:
The results are interesting and well discussed in the discussion part.
Reply: Thanks for your gratifying words.
COMMENT 2:
The table 1 data are the repetition of the data presented in the text above ( the same values as described in the text description) - I think You should decide in which form Your present the results - the text descritpion or in the table as this concerns the same.
Reply: Thanks for the suggestion that appears more appropriate and insightful for improving the readibility of all the Readers. As you requested, we have detailed our data only in the table citing throughout the text only main significant evidence, avoiding repetitive description.
COMMENT 3:
Most of the patienst were smokers - when the blood for the test was taken - in the morning? after the cigarete smoking ? before? - the condition of the blood taking in the protocol should be specified.
Reply: Thanks for the queries and suggestions. Accordingly, we have added in the Mat & Met section all the data available about protocol of blood sampling.
COMMENT 4:
In the "conclusion" part of the text You have refferences? I think this part of conclusion should be moved to the discussion summary as this concerns the literature discusssion and You should based Your conclusion on Your research (so please modify the "conclusion").
Reply: Thanks for your suggestions focused to improve readibility of our paper. According to your requests, we modified the text of the Conclusion section, moving some parts to the Discussion, avoiding reference citations and highlightening our main results according to the literature evidences.
We would like to thanks the Reviewer 1 for the comments and suggestions focused to improve the readibility of our manuscript.
Reviewer 2 Report
The authors have presented their original research assessing the oxidative and antioxidant capacity in patients with thromboangiitis obliterans (WDB) syndrome. The results have shown that patients with WDB have significantly increased TOS-TAC ratio, while also demonstrating decreased SOD when compared to control patients – all smoking. They have also shown a significant increase in coenzyme Q 10 and interestingly a decrease in MDA, while no difference in protein carbonyl.
My comments regarding the presentation of the manuscript:
Overall, the manuscript is adequately presented although there are several typographical and grammatical errors and the manuscript will benefit from line editing from a native English speaker. The discussion is a little hard to follow and would benefit from substantive editing to rearrange the order of discussion to follow a logical flow. Finally, the conclusion is unnecessarily verbose.
Regarding the scientific review of the manuscript, I found several areas where the authors need to address
- The methods written are underwhelming and provide no information as to how the sample was prepared before assaying. For example, how was the serum fractionated? Storage conditions? Was the serum diluted first in a buffer before assaying and if so what was the buffer? eet cetera. Such details are imperative even if commercial kits are utilised. In addition, there is no information as to the principle of each commercially available kit. For example, was the MDA ELISA kit based on the sandwich ELISA, or competitive ELISA? Was TAC based on Cu2+ reduction? Et cetera. The authors should elaborate on each method.
- Total cholesterol was assayed, yet there is no method to describe how this was done. Furthermore, MDA levels and CoQ10 were measured against total cholesterol though it’s not immediately clear as to why. This may be a common way to standardise certain parameters though the justification should still be described for a wider audience.
- Protein Carbonyl is a commonly used oxidative marker and in this manuscript the authors have shown no differences between WDB and control groups. Strangely, there is no discussion or mention of this outside of the results section.
- Surprisingly, MDA was significantly reduced in WDB patients. The authors have alluded to high levels of CoQ10 mitigating mitophagy, causing mitochondrial enlargement and increasing the capacity to degrade MDA. Presumably, this will be a source of ROS too. However, this does not explain why protein carbonyl is not increased, despite evidence of oxidative stress. It is incumbent upon the authors to highlight the limitations of the TAC assay employed. A review has discussed several of the limitations associated with pan TAC assays and most importantly that TAC assays excludes endogenous enzymatic activities of antioxidant systems, including catalase, and glutathione (https://doi.org/10.3109/10715761003758114). Therefore, the TAC assay is not truly indicative of the total antioxidant capacity, but rather the ‘non-enzymatic antioxidant capacity (NEAC). Under this view, it is likely that endogenous antioxidant systems such as catalase are increased in WDB patients, which could explain why protein carbonyl levels did not raise and could explain a decrease in the oxidant marker MDA. This is a common phenomena where oxidative stress does not lead to oxidative damage due to compensatory regulatory mechanisms. The authors should discuss this as a limitation to the manuscript and also as an alternative explanation to their oxidation biomarker results.
- The authors have discussed the reduction of SOD in WDB patients as mitigating the bioavailability of NO, presumably by the following pathway: O2- + NO -> ONOO. The downstream oxidation pathway typically involves nitration of tyrosine residues to form nitrotyrosine and this could be measured in the serum of patients to determine if NO was consumed by O2-. Perhaps it is worth mentioning this important pathway.
Overall, the manuscript will benefit significantly from substantive editing to address some of the concerns raised and to tidy up in typographical and grammatical errors.
Author Response
REVIEWER #2:
COMMENT 1:
Comments regarding the presentation of the manuscript:
Overall, the manuscript is adequately presented although there are several typographical and grammatical errors and the manuscript will benefit from line editing from a native English speaker. The discussion is a little hard to follow and would benefit from substantive editing to rearrange the order of discussion to follow a logical flow. Finally, the conclusion is unnecessarily verbose..
Reply: Thanks for your insightful requests, focused to improve the readibility of our MS. According to your suggestions, we corrected the typos and grammatical errors throughout the ms, improving the English editing. We have also modified the discussion rearranging the order of discussion, avoiding overall verbose and repetitive conlcusions.
COMMENT 2:
Regarding the scientific review of the manuscript, I found several areas where the authors need to address:
The methods written are underwhelming and provide no information as to how the sample was prepared before assaying. For example, how was the serum fractionated? Storage conditions? Was the serum diluted first in a buffer before assaying and if so what was the buffer? eet cetera. Such details are imperative even if commercial kits are utilised. In addition, there is no information as to the principle of each commercially available kit. For example, was the MDA ELISA kit based on the sandwich ELISA, or competitive ELISA? Was TAC based on Cu2+ reduction? Et cetera. The authors should elaborate on each method.
Reply: We would like to thank this Reviewer for the insightful requests and suggestions, surely focused to improve the readibility and significance of our ms. Accordingly, we added all the requested details of both blood collection and assay protocols, mainly in Mat & Met section.
COMMENT 3:
Total cholesterol was assayed, yet there is no method to describe how this was done. Furthermore, MDA levels and CoQ10 were measured against total cholesterol though it’s not immediately clear as to why. This may be a common way to standardise certain parameters though the justification should still be described for a wider audience
Reply: Thanks for your suggestions. We added all the explanations about your concerns, for improving readibility of the ms.
COMMENT 4:
Protein Carbonyl is a commonly used oxidative marker and in this manuscript the authors have shown no differences between WDB and control groups. Strangely, there is no discussion or mention of this outside of the results section.
Reply: Thanks for your suggestions highlightening our missing discussion about carbonyls. According to your request, we modified the text including discussions of our results on the basis of literature data.
COMMENT 5:
Surprisingly, MDA was significantly reduced in WDB patients. The authors have alluded to high levels of CoQ10 mitigating mitophagy, causing mitochondrial enlargement and increasing the capacity to degrade MDA. Presumably, this will be a source of ROS too. However, this does not explain why protein carbonyl is not increased, despite evidence of oxidative stress. It is incumbent upon the authors to highlight the limitations of the TAC assay employed. A review has discussed several of the limitations associated with pan TAC assays and most importantly that TAC assays excludes endogenous enzymatic activities of antioxidant systems, including catalase, and glutathione (https://doi.org/10.3109/10715761003758114). Therefore, the TAC assay is not truly indicative of the total antioxidant capacity, but rather the ‘non-enzymatic antioxidant capacity (NEAC). Under this view, it is likely that endogenous antioxidant systems such as catalase are increased in WDB patients, which could explain why protein carbonyl levels did not raise and could explain a decrease in the oxidant marker MDA. This is a common phenomena where oxidative stress does not lead to oxidative damage due to compensatory regulatory mechanisms. The authors should discuss this as a limitation to the manuscript and also as an alternative explanation to their oxidation biomarker results..
Reply: Thanks for your very useful and insightful suggestions. According to your requests (focused to improve the readibility and the scientific soundness of our ms), we added: 1) a paragraph of limitations of our study, 2) citation of your full sentence and improving discussion about your concerns with reference, 3) a paragraph about NEAC, 4) possible explanations and discussion with related references. THANKS again for your insightful comments: we really appreciate all of them.
COMMENT 6:
The authors have discussed the reduction of SOD in WDB patients as mitigating the bioavailability of NO, presumably by the following pathway: O2- + NO -> ONOO. The downstream oxidation pathway typically involves nitration of tyrosine residues to form nitrotyrosine and this could be measured in the serum of patients to determine if NO was consumed by O2-. Perhaps it is worth mentioning this important pathway.
Reply: Thanks for your very insightful suggestions. Accordingly, we added a paragraph mentioning and discussing the pathway you highlighted, citing linked references.
We would like to thanks the Reviewer 2 for the insightful comments and suggestions focused to improve the scientific soundness and readibility of our ms.
Round 2
Reviewer 2 Report
The authors have addressed the comments and limitations and I'm satisfied with the new state of the manuscript.